# Emerging Roles of Hedgehog Signaling in Cancer Immunity

**DOI:** 10.3390/ijms24021321

**Published:** 2023-01-10

**Authors:** Alessandro Giammona, Enrica Crivaro, Barbara Stecca

**Affiliations:** 1Core Research Laboratory, Institute for Cancer Research and Prevention (ISPRO), Viale Gaetano Pieraccini 6, 50139 Florence, Italy; 2Department of Biotechnology, Chemistry and Pharmacy, University of Siena, Via Aldo Moro 2, 50100 Siena, Italy

**Keywords:** hedgehog signaling, tumor microenvironment, immunosuppression, immune evasion, immune checkpoint inhibitors, combination therapy

## Abstract

Hedgehog–GLI (HH) signaling plays an essential role in embryogenesis and tissue homeostasis. Aberrant activation of the pathway through mutations or other mechanisms is involved in the development and progression of numerous types of cancer, including basal cell carcinoma, medulloblastoma, melanoma, breast, prostate, hepatocellular and pancreatic carcinomas. Activation of HH signaling sustains proliferation, suppresses cell death signals, enhances invasion and metastasis, deregulates cellular metabolism and promotes angiogenesis and tumor inflammation. Targeted inhibition of the HH pathway has therefore emerged as an attractive therapeutic strategy for the treatment of a wide range of cancers. Currently, the Smoothened (SMO) receptor and the downstream GLI transcriptional factors have been investigated for the development of targeted drugs. Recent studies have revealed that the HH signaling is also involved in tumor immune evasion and poor responses to cancer immunotherapy. Here we focus on the effects of HH signaling on the major cellular components of the adaptive and innate immune systems, and we present recent discoveries elucidating how the immunosuppressive function of the HH pathway is engaged by cancer cells to prevent immune surveillance. In addition, we discuss the future prospect of therapeutic options combining the HH pathway and immune checkpoint inhibitors.

## 1. Introduction

The Hedgehog–GLI (HH) signaling pathway plays key roles during embryonic development and is involved in cell proliferation, differentiation and tissue patterning. In adults, HH signaling is rapidly turned off and remains active in the stem cells of the central nervous system, skin and intestine, where it maintains tissue homeostasis and regeneration [1]. The HH signaling is aberrantly activated during the initiation and progression of a variety of cancer types, including those of the brain, skin, breast, prostate, hepatocellular and pancreatic carcinomas and hematological malignancies. The HH pathway is involved in enhancing proliferation, invasion and metastasis, in suppressing cell death signals and in deregulating the cellular metabolism [2]. Several reports have implicated HH signaling in suppressing the immune system and promoting an immunosuppressive tumor microenvironment (TME) [3]. Recent advances in cancer immunology and immunotherapy have emphasized the need for an accurate understanding of the immune-modulatory functions of oncogenic signaling pathways and their role in cancer immunity. In this review, we focus on the effects of the HH pathway on the major cellular components of the adaptive and innate immune systems and describe recent progress in elucidating how the HH pathway induces evasion from the control of the immune system. In addition, we discuss the future prospect of therapeutic options combining the HH pathway and immune checkpoint inhibitors.

## 2. Hedgehog Signaling Pathway

The HH pathway is an evolutionary signaling pathway that plays a pivotal role in patterning and organogenesis during embryonic development and in adult tissue homeostasis and repair [1]. This complex transduction pathway is coordinated by several regulatory components and post-translational modifications. In mammals, HH signaling consists of three secreted HH ligands (Sonic Hedgehog, SHH; Desert Hedgehog, DHH; and Indian Hedgehog, IHH); the 12-pass transmembrane receptor Patched 1 (PTCH1); the 7-pass transmembrane G protein-coupled receptor (GPCR) Smoothened (SMO), as the main transducer of the HH pathway; and the three zinc finger GLI transcription factors (GLI1, GLI2, GLI3), as the final mediators of the transcriptional response of HH signaling [4]. Additional members include a number of regulatory kinases [5] and Suppressor of Fused (SUFU), the main negative regulator of the GLI [6] (Figure 1).

A simplified model of HH signaling proposes that in the absence of HH ligands PTCH1 localizes to the primary cilium (PC), an organelle specialized for HH pathway transduction [7], where it suppresses the ciliary accumulation of SMO. Therefore, GLI proteins are phosphorylated by protein kinase A (PKA), casein kinase 1 (CK1) and glycogen synthase kinase-3β (GSK3β), which create binding sites for the E3 ubiquitin ligase β-transducing repeat-containing protein (β-TrCP). This promotes the complete proteasome-dependent degradation of GLI1. GLI2 and GLI3 are retained by SUFU in the cytoplasm [8,9,10,11], where they undergo partial proteasome degradation, leading to the formation of repressor forms (GLI3/2^R^) that translocate into the nucleus repressing the transcription of GLI target genes [12] (Figure 1). Degradation of β-TrCP by the endoplasmic reticulum aminopeptidase 1 (ERAP1), a key regulator of innate and adaptive immune responses [13], can protect GLI transcription factors from β-TrCP-dependent degradation and stimulate HH activity [14].

Canonical activation of HH signaling occurs upon the binding of the HH ligand to PTCH1, which exits the PC, relieving the inhibition of SMO and allowing the translocation of SMO into the PC [15]. Active SMO prevents GLI2 and GLI3 processing and promotes their dissociation from SUFU, leading to the translocation of full-length and active GLI (GLI^ACT^) into the nucleus, where they activate the transcription of GLI target genes (Figure 1). Among them, there are *GLI1* and *PTCH1*, which contribute to the creation of a positive feed-back loop. Other GLI targets include genes involved in cell proliferation (*MYC, CCND1, CCND2, FOXM1*), cell survival (*BCL-2*), angiogenesis (*ANG1/2*), epithelial-to-mesenchymal transition (*SNAIL* and *ZEB*), stemness (*NANOG* and *SOX2*) and several cytokines (*IL-6, IL-1β* and *TNF-α*) [16,17].

The HH signaling pathway is also activated through non-canonical mechanisms, which consist of the PTCH/SMO-dependent GLI-independent mechanism or in the activation of the GLI transcription factors independent of upstream PTCH/SMO. In the latter, the signal can bypass the canonical pathway to directly activate the GLI. This type of non-canonical activation occurs mainly in cancer cells and has been extensively investigated [18]. For instance, RAS-RAF-MEK-ERK1/2 and AKT signaling can regulate the nuclear localization and transcriptional activity of GLI1 in normal fibroblasts and melanoma cells [19,20,21]. In esophageal adenocarcinoma cells, the activation of mTOR signaling and S6K1 promotes the phosphorylation of GLI1 at Serine 84, preventing its association with SUFU [22]. Transforming growth factor β (TGFβ) is a strong inducer of both GLI1 and GLI2 in various human cell types, including normal fibroblasts and keratinocytes, as well as cancer cells [23]. Atypical protein kinase C ι/λ (aPKCι/λ) activates GLI1 through the phosphorylation of two residues (Ser243 and Thr304) in the zinc finger DNA binding domain of GLI1, leading to increased DNA binding and transcriptional activity [24]. The fusion oncogene Ewing Sarcoma/Friend Leukemia Integration 1 (EWS/FLI1) has been shown to induce GLI1 transcription via direct binding to the *GLI1* promoter [25]. Aside from oncogenes, loss of tumor suppressors, such as p53 or the chromatin remodeling protein SNF5, have been shown to enhance the activity of GLI1 [26,27]. Furthermore, the epigenetic modulator bromodomain-containing protein 4 (BRD4) positively regulates HH signaling by directly binding to *GLI1* and *GLI2* promoters [28], and the histone deacetylase HDAC1 can deacetylate GLI1 at Lysine 518 to promote transcriptional activation [29].

HH signaling plays a critical role in several hallmarks of cancer, such as the sustaining of proliferative signals, evasion of growth suppression and cell death and activation of invasion and metastasis, inducing angiogenesis and immune evasion [2]. Uncontrolled activation of the HH pathway is involved in a variety of cancer types. HH signaling is a key driver in the pathogenesis of basal cell carcinoma (BCC), medulloblastoma (MB) and rhabdomyosarcoma. Moreover, aberrant activation of HH signaling has been implicated in the progression of gastrointestinal, pancreatic, liver, biliary tract, ovarian, breast, prostate and lung cancers, glioblastoma, melanoma and a number of hematological malignancies [30,31].

In light of the above, a great effort has been made in the last decade to develop inhibitors targeting the HH pathway. Current inhibitors against HH signaling include SMO and GLI antagonists. These molecules have been extensively reviewed elsewhere [32,33,34,35,36,37]; hence, only the most important among them will be mentioned.

Vismodegib (GDC-0449) was the first SMO inhibitor (SMOi) to be approved in 2012 for the treatment of locally advanced and metastatic BCC [38,39,40], followed in 2015 by sonidegib (LDE225), a potent and selective SMOi with high tissue penetration and the ability to cross the blood–brain barrier [41]. In 2018 the SMOi glasdegib (PF-04449913) was approved in combination with chemotherapy for the treatment of acute myeloid leukemia patients [42]. Other SMOi are in active clinical trials, including saridegib (IPI-926) [43] and taladegib (LY-2940680), which has shown efficacy in tumors harboring the SMO-D473H mutation, which causes drug resistance to vismodegib [44].

Despite the therapeutic efficacy of SMOi, the enthusiasm for their clinical use has been hampered by the development of primary or acquired resistance, and relapse upon drug withdrawal. Notably, about 50% of BCC patients developing resistance to SMOi present mutations in *SMO*, which occur in the drug-binding pocket of SMO or in other critical domains of the transmembrane helices [45,46]. Further resistance mechanisms include *GLI2* gene amplification, and loss of the negative regulator SUFU. Inhibition of the GLI transcription factors represents an alternative strategy for the development of HH pathway inhibitors. This could be an effective approach against tumors resistant to SMOi and might have the dual advantage of blocking both the canonical and non-canonical HH pathway. To date, only a few GLI antagonists have been discovered and, except for arsenic trioxide (ATO), which is not a specific GLI inhibitor, their use has been limited to preclinical studies [33]. For instance, GANT61 and GANT58 have been shown to interfere with the binding of GLI to DNA and have shown efficacy in blocking tumor cell growth in vitro and in vivo [47]. The natural compound Glabrescione B also interferes with the interaction of GLI1/DNA and has shown therapeutic efficacy in preclinical models of HH-dependent cancers [48]. ATO, an already FDA-approved therapeutic for acute promyelocytic leukemia, has been found to suppress GLI1 transcriptional activity and block HH-induced ciliary accumulation of GLI2 [49,50]. ATO is currently in several clinical trials for cancer treatment as a single agent or in combinatorial regimen. More recently, a pharmacophore-based virtual screening approach identified quinolines and oxazino-quinoline derivatives as small molecule GLI1 inhibitors characterized by submicromolar antiproliferative activity toward human melanoma and medulloblastoma cell lines [51,52]. Further studies are in progress to optimize these small molecules and to assess their efficacy for the treatment of different types of cancer resistant to SMOi.

## 3. Hedgehog Signaling and Cancer Immunity

The innate and adaptive immune systems constitute an efficient immune surveillance machinery that recognizes and kills aberrant cells to prevent the development of cancer. The innate immune response is sustained by macrophages, neutrophils, dendritic cells (DCs), myeloid-derived suppressor cells (MDSC) and natural killer (NK) cells. An adaptive immune response comprises CD4+ helper T lymphocytes, CD8+ cytotoxic T lymphocytes (CTLs) and B lymphocytes. Immune cells and cancer cells also interact with stromal cells, particularly cancer-associated fibroblasts (CAF), which contribute to tumor growth and metastasis [53,54]. Cross-talk between cancer cells and neighboring immune cells is mediated by a variety of signaling pathways and cytokines, and ultimately results in a microenvironment that stimulates tumor growth and metastasis. It is essential to understand the nature of these reciprocal communications to design novel therapeutic approaches that simultaneously target multiple components of the TME. The administration of immune checkpoint inhibitors has shown impressive therapeutic efficacy in several types of cancer [55]. It is fair to predict that combinatorial treatments targeting HH signaling and immunosuppressive mechanisms might improve the therapeutic response in patients with cancers dependent on HH pathway activation.

In the following paragraphs, we will summarize recent discoveries reporting the impact of HH signaling in the context of tumor immunity, focusing on the effects exerted on major cellular components of the innate and adaptive immune systems, CAF and immune checkpoint molecules.

### 3.1. HH Signaling and T Lymphocytes

T lymphocytes are the most effective mediators of an adaptive anti-tumor response. The cytotoxic CD8+ T cell population, together with CD4+ T helper (Th1) cells through the production of IL2 and INFγ, recognize tumor cells presented by antigen-presenting cells (APC). During the lysis process, target cells are linked by activated cytotoxic T lymphocytes (CTLs), which release cytotoxic granules containing perforin and granzymes, leading to target cell death. On the contrary, CD4+ T cell subset Th2 orchestrates an immunosuppressive phenotype [56,57]. T lymphocytes control tumor progression by infiltrating the TME. In fact, T cell abundance, functional activity and spatial distribution in the TME represent important prognostic and predictive factors for immune checkpoint inhibitors [58,59,60].

SHH signaling is not only important during thymocyte differentiation, but also for T-cell development and activation [61]. A study from de la Roche and colleagues elucidated the role of HH signaling in the immunological synapse during T-cell activation. Activation of the T cell receptor (TCR) in CTL triggers HH signaling, which, in turn, increases levels of the GTPase RAC1, promoting centrosome polarization, actin remodeling, granule release and target cell killing. HH signaling is required for CTL killing and centrosome polarization to the immunological synapse; indeed, pharmacological inhibition of SMO or GLI1 led to the functional disruption of the immunological synapse and loss of T-cell effector activity [62]. It remains to be investigated whether the administration of SMO inhibitors affects cytotoxic T-cell functions in patients.

Peripheral T-cell activation is initiated by the interaction of TCR with its major histocompatibility complex (MHC)–peptide ligand. For full activation, T-cells require co-stimulation by binding CD28 to CD80 and CD86 on APCs. TCR and CD28 ligation leads to a number of TCR-proximal phosphorylation events, the release of intracellular Ca^2+^ and to the activation of key transcription factors including the activator protein 1 (AP-1) complex, members of the nuclear factor of activated T-cells (NFAT) family and nuclear factor κB (NF-κB) [63]. Using a transgenic mouse model in which an activator form of GLI2 (GLI2^ACT^) was expressed in the T-lineage, Furmanski and colleagues demonstrated that GLI2^ACT^ reduces T-cell activation and proliferation following TCR activation [64]. Mechanistically, expression of GLI2^ACT^ in the T-cells altered gene expression profiles, impaired the TCR-induced Ca^2+^ flux and nuclear expression of NFAT2 and attenuated signaling pathways upstream of the activator protein-1 (AP-1) and nuclear factor kappa B (NF-κB) complexes, leading to the reduced activation of these important transcription factors. In contrast, the inhibition of HH signaling by a repressor form of GLI2 (GLI2^R^) led to an increase in NF-κB activity upon TCR ligation and a change in the molecular composition of AP-1 [64]. These findings hold important implications for understanding the immune regulation in tissues that express ligands able to activate GLI-dependent transcription.

In the context of allergic diseases, activation of HH signaling can exert different effects and outcomes depending on the tissue. In the lung, SHH signals to T-cells to promote Th2 differentiation driving allergic asthma, so that lowering HH pathway activation ameliorates allergic disease. For instance, the ligand SHH is upregulated in the airway of mice with allergic airway disease (AAD) and the expression of an activator form of GLI2 (GLI2^ACT^) in T-cells directly increases the production of IL-4, promoting the differentiation of naïve T cells to a Th2 phenotype and exacerbating allergic responses [65]. Another study from the same group showed that GLI-dependent transcription is activated in T cells in vivo during murine AAD, a model for the immunopathology of asthma, and that genetic repression of GLI signaling in T cells decreases the differentiation and recruitment of Th2 cells to the lung [66]. Likewise, the systemic inhibition of SMO in a papain-induced mouse model of allergic airway disease lowered lung T-cell infiltrates, ameliorated Th2 inflammation and reduced the expression of the Mucin gene Muc5ac and serum IgE [67]. On the contrary, in skin, SHH appears to induce regulatory T-cell function and therefore its upregulation is protective against inflammation and disease, and SMO inhibition aggravates it. Indeed, HH signaling has been shown to improve the disease in the context of atopic dermatitis (AD) through GLI2-driven immune regulation and to induce the differentiation of immunosuppressive Treg cells expressing elevated levels of FOXP3 and TGFβ. Consistently, the inhibition of the HH pathway with the SMO inhibitor PF-04449913 promoted skin inflammation and chronic AD in vivo [68]. It remains unclear why HH signaling affects T-cells differently in lung and skin in the context of allergic diseases. Plausible explanations for these differences might be due to multiple signals that T cells receive in each environment (lung versus skin) or intracellular differences between T-cells in the different tissues when HH signaling is activated. In conclusion, several studies have suggested that targeting HH signaling might be a useful therapeutic approach to prevent or reduce allergic airway inflammation [65,66,67]. However, given the tissue-dependent differences in outcome of inhibiting HH signaling in allergic disease of skin and lung, and the fact that some individuals may have several different sites of allergic inflammation, great caution must be taken before treating with SMO inhibitors patients with allergic disease of the skin.

In the context of cancer, activation of HH signaling has been reported to suppress CD8+ T cell recruitment, whereas its pharmacological inhibition promotes the infiltration of CD8+ T cells in several cancer models. An important study from Otsuka and collaborators investigated the effects of SMOi on the immune response in 23 BCC patients [69]. Treatment with the SMOi vismodegib (22 patients) or sonidegib (one patient) for four weeks led to changes in the TME, characterized by increased levels of MHC-I expression in cancer cells and increased infiltration of CD4+ and CD8+ cells in the tumor. These changes were associated with the upregulation of local cytokines CCL18, CCL21 and CXCL9, which are thought to have a critical role in tumor suppression [69]. A recent study in a mouse model of mammary carcinoma showed that HH pathway inhibition with vismodegib remodels the gut microbiota and increases the proliferation of resident CD8+ T cells across the immune network in the colon [70] (Figure 2). Other examples of the negative impact of HH signaling on CD4+ and CD8+ T in the TME are reported below.

### 3.2. Regulation of Tumor-Associated Macrophage Behavior by HH Signaling

Tumor-associated macrophages are classified into two major phenotypes: inflammatory M1 (classically activated) or immune-suppressive M2 (alternatively activated) [71]. Macrophages are the most abundant immune population in the TME and can account for 50% of the tumor mass. M2 macrophages sustain cancer cell proliferation and invasion. In turn, tumor cells secrete cytokines that influence TAM to switch to an M2 phenotype, which is associated with poor clinical outcomes in several cancer types [72].

Mounting evidence indicates that one of the main immunosuppressive roles of the HH pathway consists of the alternative activation of macrophages to M2 phenotype (Figure 2). Using an orthotopic breast cancer mouse model, Hanna and colleagues [73] demonstrated that the inhibition of HH signaling with the SMOi vismodegib significantly reduced M2 macrophages, MDSCs and Treg cells in the primary tumor without affecting tumor growth, while it boosted the number of M1 macrophages, cytotoxic CD8+ T-cells and dendritic cells, decreasing pulmonary metastasis. The process of the acquisition of an alternative activated M2 macrophage phenotype is characterized by the increased expression of the M2 markers Arg1 and Cd206, associated with elevated levels of Gli1. Exogenous SHH further potentiated the expression of Arg1 and Cd206, increasing the expression of Gli1, whereas the pharmacological inhibition of SMO or GLI inhibited alternative polarization of macrophages. Furthermore, inhibition of HH pathway in immunosuppressive M2 macrophages enables their conversion to an inflammatory phenotype. Notably, macrophage depletion using liposomal clodronate further improved the therapeutic efficacy of HH pathway inhibition, eliciting pro-inflammatory and immunogenic phenotypes. Combination of liposomal clodronate with vismodegib resulted in a significant reduction in pro-tumorigenic M2 macrophages, MDSCs, helper T cells type 2 (Th2) and Treg cells, and the concomitant increase in inflammatory M1 macrophages, dendritic cells, cytotoxic T cells and Th1 cells [73].

Another report confirmed the essential role of HH signaling in promoting M2 polarization of TAMs [74]. Using several murine tumor models, the authors showed that tumor cells secrete the ligand SHH, which is critical for TAM M2 polarization. Mechanistically, the authors demonstrated that the HH-induced polarization in TAMs suppresses CD8+ T cell recruitment to the TME through the inhibition of cytokines CXCL9 and CXCL10. Mechanistically, HH-induced TAM M2 polarization and immunosuppressive function are mediated by the Kruppel-like factor 4 (KLF4), which is transcriptionally regulated by GLI1 in macrophages (Figure 2). Notably, vismodegib and the anti-PD1 antibody have synergistic anti-tumor effects in immunocompetent hepatocarcinoma and lung carcinoma xenograft mouse models [74].

A recent report established a novel role for HH signaling in regulating a complex metabolic network in mammary TAM [75]. M1 and M2 macrophages have distinct metabolic circuitries that contribute to their survival and different functions in the immune response. Anti-tumorigenic M1 macrophages rely mainly on glycolysis, while M2 macrophages use oxidative phosphorylation to produce the energy required for their tumor-promoting functions [76]. Using two immunocompetent models of mammary tumors, Hinshaw and colleagues discovered that vismodegib induces alterations in metabolic processes, including metabolic sensing, mitochondrial adaptations and lipid metabolism. More specifically, HH pathway inhibition in M2 macrophages decreases flux through the uridine diphosphate N-acetylglucosamine (UDP-GlcNAc) biosynthesis pathway, diminishing the immune-suppressive phenotype of M2 macrophages, which shifts their metabolism from fatty acid oxidation to glycolysis (Figure 2). These findings reported a novel immune–metabolic function of HH signaling that could be clinically exploited to promote an immunogenic response to cancer [75].

Finally, it was shown that the activity of the three GLI transcription factors modulates the infiltration of macrophages in animal models of pancreatic ductal adenocarcinoma (PDA) [77]. The authors showed that Gli1, Gli2 and Gli3 are expressed in the healthy pancreas, and expand throughout PDA progression. The genetic depletion of Gli2 and Gli3 in fibroblasts at the pre-cancerous stages reduces the infiltration of immunosuppressive macrophages and increases the infiltration of T cells. On the contrary, the combined ablation of Gli1/Gli2/Gli3 promotes macrophage infiltration and exclusion of T cells [77]. These findings demonstrate that the activity of all three GLI transcription factors regulates immune infiltration.

### 3.3. Suppression of Natural Killer Cells by HH Signaling

Natural killer (NK) cells are circulatory, innate lymphoid cells known for their cytotoxic function. NK cells are very efficient at eliminating malignant cells and restraining metastasis. NK cells use death receptor-mediated apoptosis and perforin/granzyme-mediated cytotoxicity to target cancer cells and prevent tumor growth [78]. It is important to mention that NKs are less efficient at killing tumor cells in the TME, due to several mechanisms that cancer cells employ to evade destruction by NK cells [79].

In an invasive model of pancreatic ductal adenocarcinomas (PDA), the ablation of Gli2 and Gli3 in fibroblasts was shown to decrease tumor growth by recruiting NK cells during the late stages of tumorigenesis. Subcutaneous injection of a murine pancreatic ductal adenocarcinoma cell line (KPC) with mouse Gli2/Gli3-knock-out (KO) fibroblasts in mice led to a reduction in tumor growth, decreased the recruitment of MDSCs and increased NK cells (Figure 2). This effect on immune cells is specific to Gli2/Gli3 KO fibroblasts, since Gli1/Gli2/Gli3 KO fibroblasts do not impact the infiltration of NK and MDSCs. Notably, the depletion of NK cells in tumors co-injected with Gli2/Gli3 KO fibroblasts rescues tumor growth, suggesting that the loss of Gli2/Gli3 in fibroblasts restrains tumor growth through the recruitment of NK cells [77].

### 3.4. Myeloid-Derived Suppressor Cells

MDSCs consist of a heterogenous cell population with myeloid origin, including myeloid progenitor cells and immature macrophages, immature granulocytes and immature dendritic cells [80]. MDSCs in the TME suppress innate and adaptive immune responses. Moreover, MDSCs can initiate the formation of the premetastatic niche, enhance stemness and angiogenesis and promote the metastatic process by inducing EMT through the secretion of IL-6 [81].

One of the first hints revealing the immunosuppressive function of HH signaling in the TME came from a study showing that the overexpression of a constitutive active form of Smo (SmoM2) in a mouse model of BCC is associated with the accumulation of immunosuppressive MDSC in BCC lesions [82]. The transforming growth factor β (TGFβ)-CCL2 axis is responsible for the recruitment of MDSC in SmoM2-induced BCC lesions (Figure 2). Consistent with these findings, the pharmacologic inhibition of the CCL2 receptor, expressed by MDSCs, reduced MDSC recruitment and HH-driven BCC development in mice [82].

### 3.5. Regulatory T Cells

Regulatory T cells (Treg) are characterized by the expression of the transcription factor FOXP3. In the TME, they play a pivotal role in cancer immune evasion by suppressing the antitumoral immune response through different mechanisms, including the production of growth factor and cytokines, such as TGFβ and IL-10 [83].

A recent report showed that the activation of HH signaling in epidermal cells induces an immunosuppressive TME in a mouse model of BCC. Activation of HH signaling promotes the strong accumulation of immunosuppressive regulatory Tregs, which are localized in intra- and peri-tumoral regions, suggesting a possible role of Tregs in the immunosuppression of the BCC microenvironment (Figure 2). BCC lesions also presented a pronounced infiltration of neutrophils, consistent with elevated levels of chemokines, such as CCL2 and CCL3, two potent chemoattractants for myeloid cells [84].

Besides the effects of HH on immunosuppression in cancer, HH-induced Treg formation can also restrain inflammation-driven diseases. For instance, a study from Lee and colleagues reported that the genetic or pharmacologic inhibition of the HH pathway worsens colon inflammation (colitis) and promotes colitis-associated cancer development in mice. Conversely, the activation of the HH pathway ameliorates colitis and restrains the initiation and progression of colitis-induced adenocarcinoma. The authors found that HH pathway stimulation exerts its effects through the increased expression of the anti-inflammatory cytokine IL-10 in HH pathway-responsive stromal cells and concomitant increases in CD4+ FOXP3+ regulatory T cells in the colon [85]. These findings could have important consequences for cancer patients receiving systemic SMO antagonists. Treatment with SMO antagonists may increase the risk of contracting severe colitis. Furthermore, these results may explain the failure of colon cancer trials using HH pathway inhibitors, given that it is plausible that the pro-inflammatory responses to SMO-targeting may contribute to the acceleration of cancer progression, forcing the termination of the clinical studies [86,87].

### 3.6. Cancer-Associated Fibroblasts

Cancer cells and immune cells in the TME interact with stromal cells, including cancer-associated fibroblasts (CAF). CAFs have been implicated in tumor proliferation, invasion and metastasis. CAFs can secrete immunosuppressive cytokines that polarize macrophages to the M2 phenotype and contribute to the exhaustion of CD8+ T-cell [88].

Steele and colleagues found that the inhibition of the HH pathway alters the composition of CAFs and immune infiltration in the pancreatic TME. The authors demonstrated that HH pathway activation is higher in myofibroblastic CAFs (myCAF) compared to inflammatory CAFs (iCAF) in both mouse and human PDA. Notably, HH pathway inhibition with the SMOi LDE225 impaired PDA growth in orthotopic and genetic engineered mouse models of PDA. However, LDE225 treatment alters the ratio of myCAFs and iCAFs in PDA, favoring the increase in iCAF subpopulation, decreases cytotoxic T cells and increases regulatory T cells, which results in an immunosuppressive TME [89] (Figure 2). The authors speculated that the detrimental effects associated with long-term HH pathway inhibition may depend on the enrichment of potentially tumor-promoting iCAFs rather than the depletion of the tumor-restraining myCAF population [89]. The heterogeneous response of CAF populations in PDA TME to HH pathway inhibition might explain the disappointing outcome of clinical trials targeting SMO in PDA patients [90,91].

### 3.7. Induction of Immune Checkpoint Molecules by HH Signaling

Another mechanism by which HH signaling promotes immunosuppression is by inducing the expression of immune checkpoint molecules. One of the most critical checkpoint pathways is mediated by the programmed cell death protein 1 (PD1) and its ligand, programmed death ligand 1 (PDL1). PD1 is highly expressed by activated T cells, B cells, DCs and NK cells, whereas PDL1 is expressed in several types of tumor cells. The PD1/PDL1 interaction inhibits CTL effector function, driving immune evasion and cancer cell proliferation [92].

Several studies have indicated that HH signaling regulates PDL1 expression in tumor cells and TME (Figure 2). For instance, it has been shown that HH signaling induces PDL1 expression in human pancreatic ductal adenocarcinoma, gallbladder cancer and small cell lung cancer cells under hypoxic conditions. Consistently, the inhibition of the HH pathway decreases PDL1 expression, increasing CD8+ lymphocyte activation [93]. Likewise, Chakrabarti and colleagues showed that GANT61 treatment reduces PDL1 expression and tumor cell proliferation in gastric cancer organoids derived from GLI2-expressing mice. Notably, autologous cultures of GLI2-expressing gastric organoids, dendritic cells and CTL treated with anti-PDL1 neutralizing antibody resulted in cytotoxic T cell-induced tumor apoptosis [94].

Activation of HH signaling in mouse models of BCC increased the expression of immune checkpoint molecules, including PD1/PDL1. Surprisingly, anti-PD1 monotherapy did not appear to reduce tumor growth [84]. In the context of BCC, it will be interesting to investigate whether the combination of anti-PD1 with HH pathway inhibitors might affect tumor growth.

A recent report by Petty and collaborators revealed the key role of SHH-dependent PDL1 upregulation in TAMs in suppressing antitumor immunity [95]. Using a myeloid-specific Pdl1-knockout mouse model, the authors demonstrated that the deletion of Pdl1 in TAMs rescues intratumor CD8+ T cell function and suppresses tumor growth, providing evidence for the critical role of TAM-derived PDL1 in suppressing intratumor CD8+ T cell function. The authors further showed that tumor-derived SHH, through STAT3 signaling, induces PDL1 expression on M2 TAMs to suppress tumor-infiltrating CD8+ T cells, resulting in enhanced tumor progression [95]. These findings provide important insights for the development of novel therapeutic strategies for the treatment of hepatocellular carcinoma (HCC) and other SHH-expressing human cancers.

PDL1 upregulation was also observed in some cases of medulloblastomas, where the highest PDL1 expression was found in a patient with SHH subtype MB [96].

## 4. Combining HH Pathway Inhibitors with Immune Check Point Inhibitors

Given the predominant immunosuppressive role of HH signaling in a variety of cancer types, HH pathway antagonists are predicted to synergize with immune checkpoint inhibitors (ICI) and this combinatorial targeted and immune therapy might hold great promise in the fight against cancer.

Through in silico analysis it was shown that the presence of mutations in *PTCH1* might be a potential biomarker for predicting the response of colorectal cancer patients to ICI. *PTCH1*-mutated tumors present higher proportions of CD8 + T cells, activated NK cells and M1 type macrophage infiltration. Patients with *PTCH1* mutations have better progression-free survival, overall survival and are associated with better prognosis [97]. Future investigations in larger clinical cohorts are warranted to confirm these potential interesting correlations.

Another study using transcriptional data and clinical outcomes from across 14 cancer types obtained from genome atlases investigated the role of HH in the TME, exploring its potential as a negative biomarker for immune checkpoint inhibitor therapy. A single biomarker strategy is not accurate enough to identify the patients who could benefit from such a strategy or treatment. The authors demonstrated by single-sample Gene Set Enrichment Analysis (ssGSEA) on different and independent cancer patient cohorts the need to apply a joint prediction strategy. Thus, they developed one which combined HH signaling with PDL1 expression that seems to be reliable for the resistance to ICI prediction within high PDL1 expression patients [98].

A recent study from Lo Cascio and colleagues highlighted the use of a combination of HH signaling inhibitor and anti-PDL1 immunotherapy to improve the clinical outcome of ovarian cancer. Cancer-associated mesenchymal stem cells (CA-MSCs) are a critical driver of the immune-suppressive TME in ovarian cancer. Using an immune “hot” mouse ovarian cancer model, the authors discovered that the inhibition of HH signaling with the SMOi IPI-926 reverses CA-MSC–driven tumor immune exclusion and restores response to anti-PDL1 therapy. In particular, HH pathway inhibition is able to reduce the number of tumor-associated monocytes and macrophages, reverse the CD8+ T cell tumor immune exclusion and increase the influx of NK cells into the TME, rendering the tumor responsive to anti-PDL1 treatment [99].

A report from Petty and colleagues showed that the combination of vismodegib with the anti-PD1 antibody resulted in a synergistic decrease in liver tumor in mice (Hepa1-6 and LLC-1 tumors) by reversing the phenotype of TAM from M2 to M1 and increasing recruitment of CD8+ T-cells into the TME [74]. A small size clinical trial (NCT02690948) showed a 44% vs. 29% overall response rate between pembrolizumab-treated patients (n = 9) and patients treated with pembrolizumab + vismodegib (n = 7). These results are discouraging because they suggest that immunotherapy might work better than the combination, although one-year progression-free survival was more favorable to combination therapy (62% vs. 83%) [100].

Notably, results from a phase 2 clinical trial (NCT03132636) showed that the PD1 antibody cemiplimab is an active treatment option for patients with locally advanced basal cell carcinoma who had progressed on or are intolerant to SMOi therapy (vismodegib, sonidegib or their combination). The safety profile of cemiplimab in this study was also acceptable. This is the first study to show the activity of a systemic therapy in locally advanced basal cell carcinoma after HH pathway inhibitor therapy [101].

A phase 2 study is investigating nivolumab (NIVO) alone or plus ipilimumab (IPI) for patients with locally advanced unresectable (laBCC) or metastatic basal cell carcinoma (mBCC). The purpose of this study is to evaluate the efficacy of NIVO (anti-PD1) alone and in combination with IPI (anti-CTLA-4) in patients with laBCC or mBCC, either in the first line setting or after HH pathway inhibitors (https://clinicaltrials.gov/trial identifier (accessed on 25 March 2022) NCT03521830). The trial is still recruiting, and no results are available yet.

The outcome of these ongoing clinical trials with immune checkpoint inhibitors for the treatment of metastatic or unresectable BCC alone or in combination with SMO inhibitors will inform whether immunotherapy or combinatorial treatments can increase the efficacy and durability of the response in BCC patients.

## 5. Conclusions

It is clear from the presented literature that HH signaling has different effects on the immune microenvironment of malignant and non-malignant tissues, and also in the context of allergic diseases it can exert surprisingly divergent outcomes. For instance, in the lung, HH signaling promotes the differentiation of T cells to a Th2 phenotype exacerbating allergic responses. By contrast, in the skin it induces T-cell function and therefore its upregulation is protective against inflammation, whereas its inhibition aggravates the disease [65,66,67,68].

In this review we have described the snapshots of HH pathway effects on tumor immunity, illustrating how different immune cell types present in the TME react to HH signaling modulation in a variety of experimental systems. In most cases, findings in tumor models have converged on the immune suppressive role of HH signaling [69,70,73,74,75,77,82,84,89]. However, there are several questions that remain unanswered. For instance, in a patient tumor it is unclear whether immune cell responses occur in concert or sequentially during tumor progression, and, in the latter case, in which order. It will be also critical to interrogate genomic studies, in particular single-cell profiling, and possibly address the spatio-temporal responses of HH-mediated immune evasion. Understanding the precise molecular basis of how aberrant activation of HH signaling affects anti-tumor immunity and the exact cellular subtypes will support the development of more effective cancer therapies. On this point, the synergisms observed between HH inhibition and checkpoint blockade in some cancer types may reflect a hierarchy of immune-suppressive events, where the activation of HH signaling decreases the initial immune response by reducing the ability of T cells to infiltrate the tumor, whereas PD1/PDL1 signaling is engaged at later stages by repressing T cell effector functions.

Given the immune modulatory role of HH inhibitors and the finding that the administration of SMO inhibitors leads to the loss of T-cell effector activity [62], the use of HH inhibitors in combination with immunotherapy is not without its challenges. At the moment it is unknown whether the administration of SMO inhibitors affects cytotoxic T-cell functions in patients. Therefore, future studies will need to take into consideration the possible negative impact of HH targeting on the antitumoral response, particularly those involving immune checkpoint inhibitors. A deeper understanding of the effect of HH pathway activation and inhibition on the immune response is crucial and this knowledge will be essential to devise safe and effective therapies combining HH and immune checkpoint inhibitors.

## Figures and Tables

**Figure 1 ijms-24-01321-f001:**
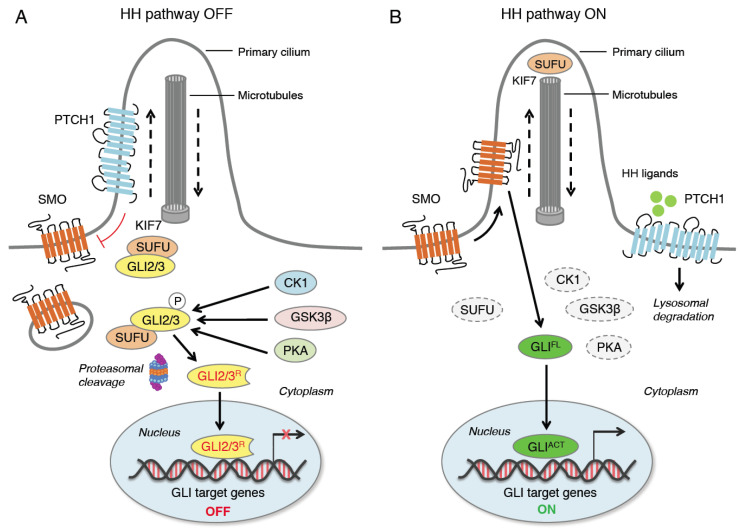
Canonical activation of HH signaling. When HH ligands are not present (**A**), PTCH1 represses SMO by preventing its entry into the primary cilium (PC). GLI2 and GLI3 are sequestered in the cytoplasm by SUFU and phosphorylated by PKA, CK1 and GSK3β. The GLI undergo ubiquitination through the E3 ubiquitin ligase β-TrCP. GLI1 is fully degraded, whereas GLI3 and, to a lesser extent, GLI2 undergo partial proteasome degradation, leading to the formation of repressor forms (GLI3/2^R^) that move into the nucleus inhibiting the transcription of GLI target genes. In the presence of HH ligands (**B**), PTCH1 is displaced from the PC and undergoes lysosomal degradation, and SMO translocates into the PC. Active SMO relieves the SUFU-mediated suppression of GLI2 and GLI3, triggering a signaling cascade that leads to the translocation of full length activated forms of GLI (GLI^ACT^) into the nucleus, where they promote the transcription of GLI target genes. KIF7 is a kinesin protein that acts in anterograde transport (from base to tip) of the PC. CK1, casein kinase 1; GLI2/3^R^, GLI2/3 repressors; GLI^ACT^, GLI activators; GLI^FL^, GLI full length; GSK3β, glycogen synthase kinase 3β; HH, Hedgehog; KIF7, kinesin family member 7; PKA, protein kinase A; PTCH1, Patched 1; SMO, Smoothened; SUFU, Suppressor of Fused; β-TrCP, β-transducin repeat-containing protein.

**Figure 2 ijms-24-01321-f002:**
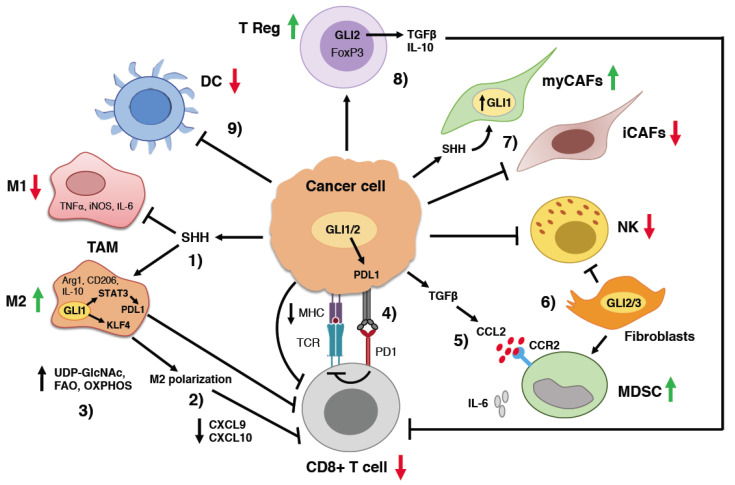
Impact of HH signaling on the immune tumor microenvironment. The HH pathway is active in cancer cells and in several other cell types present in the tumor microenvironment, including TAM, T Reg and fibroblasts. (1) Cancer cells release SHH, driving the polarization of M2 macrophages and repressing M1 macrophages. M2 polarization is mediated by KLF4, which is transcriptionally regulated by GLI1. (2) HH-induced polarization of TAMs suppresses CD8+ T cell recruitment through the inhibition of CXCL9 and CXCL10. (3) HH signaling regulates the immunosuppressive metabolism in M2 TAMs by increasing the UDP-GlcNAc (uridine diphosphate-N-acetylglucosamine) biosynthesis pathway, fatty acid oxidation (FAO) and oxidative phosphorylation (OXPHOS). (4) HH-induced PDL1 expression in cancer cells inhibits tumor-specific CD8+ T cells via binding to PD1. (5) HH-induced TGFβ activates the CCL2/CCR2 axis to recruit immunosuppressive MDSCs in BCC. (6) Fibroblast-specific ablation of Gli2/Gli3 decreases the recruitment of MDSCs and increases NK cells suppressing tumor growth in PDA. (7) Cancer-secreted SHH activates HH signaling in surrounding myCAFs in PDA. (8) GLI2 drives the production of immunosuppressive cytokines and growth factors (IL-10 and TGFβ) which results in the inactivation of tumor-specific CD8+ T cells. (9) HH activation decreases the recruitment of DCs.

## Data Availability

Not applicable.

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
