# Peer review of "Emerging Roles of Hedgehog Signaling in Cancer Immunity"

_ijms, 2023, doi:10.3390/ijms24021321_

Round 1

Reviewer 1 Report

Dear authors,

I would like to inform that after few rounds of reviewing, I am impressed by the work from the authors. The manuscript has been written well, the flow of ideas information and ideas are acceptable. The manuscript have scientific merit for publication. Therefore, I would like to suggest to accept in the present form.

Thank you. 

Author Response

We thank the Reviewer for his/her very positive comments on our manuscript.

Reviewer 2 Report

In their work, the authors described very comprehensively the effects of Hedgehog-GLI (HH) signaling on cellular components of the adaptive and innate immune systems. They have also presented recent discoveries elucidating how the function of HH is engaged by cancer cells and discussed the link between the HH pathway and immune checkpoint inhibitors. The review is well-written, although before it is published, I have a few minor suggestions for the authors:

1. I suggest a more detailed presentation of HH canonical activation in figure 1 and the text below to make the chapter and its auxiliary scheme more accessible to the reader. If the SMO directly affects the components of the signaling cascade, the figure should also clearly indicate that. 

2. All chapter 3, "Hedgehog signaling and cancer immunity," refers the reader to figure 2, which could be more informative in emphasizing and enabling to understand one of the main goals of the review paper, i.e., HH signaling in the context of tumor immunity and major cellular components of the two immune systems molecules. Also, the title suggests a distinction between adaptive and innate immune responses under the influence of Hedgehog signaling, while the figure inadequately and schematically depicts the cancer cells while HH signaling is "on" or "off ." I suggest changing it to the way it is based more on what is described in chapter 3. In addition, there is no need to mark out "see text for details."

Apart from these minor remarks and suggestions, I have no other comments about the work. After introducing the proposed changes, I will be very pleased to recommend the review for publication in the International Journal of Molecular Sciences. 

Author Response

We thank the Reviewer for his/her positive comments on our manuscript.

As suggested, we made a more detailed representation of canonical HH signaling in Figure 1 and made legend more accessible to the reader (point 1).

We also modified Figure 2 to better depict the key processes regulated by HH signaling in the major cellular components of the adaptive and innate immune systems, cancer associated fibroblasts (CAFs) and immune checkpoint molecules, as described in chapter 3 (point 2).

We also corrected few typos throughout the manuscript.

Reviewer 3 Report

The author has presented significant research addressing the future prospect of therapeutic options combining the Hedgehog pathway and immune checkpoint inhibitors. All the sections of reviews are highly correlated and relevant to the theme of the review.

The abstract is very well framed and presented the theme of the manuscript in a concise effective manner. The introduction section has presented a very good description of the hedgehog signaling pathway along with the immune checkpoint inhibitors.

Diagrams are highly effective and correlated to the sections of the manuscript. Immune checkpoint inhibitors have been gaining wider attention nowadays in the field of cancer therapeutics. The authors have also included this section along with the hedgehog signaling pathway.

The conclusion section is very well framed and the author has summarised the findings in a very appropriate manner. I would recommend the author to accept this manuscript in its current form.

Author Response

Response: We thank the Reviewer for his/her very positive comments on our manuscript. We also corrected few typos throughout the manuscript.